# Development of a New Robust Stable Walking Algorithm for a Humanoid Robot Using Deep Reinforcement Learning with Multi-Sensor Data Fusion

Çağrı Kaymak [1] , Ayşegül Uçar [1,*] and Cüneyt Güzeliş [2]

1  Mechatronics Engineering Department, Engineering Faculty, Fırat University, Elazig 23119, Turkey
2  Electrical and Engineering Department, Engineering Faculty, Yaşar University, Izmir 35100, Turkey
*  Correspondence: agulucar@firat.edu.tr

**Abstract:** The difficult task of creating reliable mobility for humanoid robots has been studied for decades. Even though several different walking strategies have been put forth and walking performance has substantially increased, stability still needs to catch up to expectations. Applications for Reinforcement Learning (RL) techniques are constrained by low convergence and ineffective training. This paper develops a new robust and efficient framework based on the Robotis-OP2 humanoid robot combined with a typical trajectory-generating controller and Deep Reinforcement Learning (DRL) to overcome these limitations. This framework consists of optimizing the walking trajectory parameters and posture balancing system. Multi-sensors of the robot are used for parameter optimization. Walking parameters are optimized using the Dueling Double Deep Q Network (D3QN), one of the DRL algorithms, in the Webots simulator. The hip strategy is adopted for the posture balancing system. Experimental studies are carried out in both simulation and real environments with the proposed framework and Robotis-OP2's walking algorithm. Experimental results show that the robot performs more stable walking with the proposed framework than Robotis-OP2's walking algorithm. It is thought that the proposed framework will be beneficial for researchers studying in the field of humanoid robot locomotion.

**Keywords:** humanoid robot; stable walking; parameter optimization; Deep Reinforcement Learning; multi-sensor

## 1. Introduction

Today, robots are used in many areas, from automation systems to the defense industry. Artificial intelligence and learning algorithms are subject to continuous development and improvement so that robots can efficiently perform the daily tasks of humans. Thus, robotics has become an essential element and has found a place in many different actions in daily life.

Robotic systems can be classified into two main areas: manipulator robots with a fixed workspace and mobile robots with a portable workspace. Although the wheeled robots can travel quite quickly, they can only go across flat ground. While slower than wheeled robots, tracked robots can go across more difficult terrain. In challenging terrain, legged robots outperform wheeled and tracked robots, as they show greater mobility and flexibility, are highly adaptable to terrain differences, and cause less environmental damage. Considering this superiority and the fact that approximately 80% of the earth is inaccessible by conventional wheeled vehicles, legged robots are more prominent in the field of mobile robots [1].

Humanoid robots are two-legged robots made with a human-like design and have different functions, such as being able to move and speak. They have an important place among mobile robots because of their similarity to humans, ability to move in environments suitable for humans, and ability to use tools designed for humans.

A humanoid robot's design focuses primarily on balance control. A simple way to control the balance of a bipedal walking robot is to mimic the locomotion of humans, which is typically human. An average human gait consists of all the movements that occur with the forward orientation of the Center of Gravity (CoG) through the limbs and body. During walking, displacements and load are carried out by the legs, while, by adjusting to the CoG changes brought on by the movement of the legs, the remainder of the body moves on.

In biomechanical research, understanding the bipedal locomotion stability and gait mechanism is essential for better understanding how humans move from one place to another [2]. Although it may seem simple, human locomotion is a highly complex state involving multiple Degrees of Freedom (DoF) combined with the complex nonlinear dynamics produced due to the various extensor and flexor muscle groups in the lower body. While humanoid robots are known for their ease and flexibility to move around a wide range of terrain, the main concern is stability as they have a high-dimensional and nonlinear system. Over the years, many control scientists have developed an interest in finding a solution to the stability issue with the bipedal gait system [3].

There are approaches to the development of stable walking when recently proposed gait frameworks are reviewed. The dynamic model of a robot, for which the gait planner and controller are created, forms the basis of the fundamental framework. Some restrictions are taken into account in this framework to lessen the difficulty of creating a whole-body dynamics model Another framework's primary component is a network of signal generators that together produce intrinsically rhythmic signals [4,5]. The Central Pattern Generator (CPG) is the basis for this kind of framework. It draws inspiration from research on the neurophysiology of vertebrate and invertebrate species [6,7]. These studies have demonstrated that CPGs in the spinal cord that are coupled in a specific way are responsible for rhythmic locomotion, including walking, running, and swimming. Oscillators are often assigned to each link in this kind of framework to produce set points (torque, position, etc.). Many humanoid robots have more than 20 DoF. As a result, adjusting the oscillators' parameters is both challenging and labor-intensive in terms of experimentation. Robots' knowledge is typically static in these two types of frameworks mentioned above. It does not improve from past experience. For this reason, they must at least reconfigure the parameters to adapt to new environments. According to another framework, based on Reinforcement Learning (RL), walking trajectories are generated [8]. According to this framework, walking trajectories are generated following training that needs a lot of samples and takes a long time. The framework attempts to learn how to generate walking trajectories depending on a function during training. The methodologies outlined above are combined to create the final framework [9–13]. This kind of framework is also referred to as a hybrid walking framework. It seeks to maximize performance by utilizing the various strengths of each strategy.

Traditional control theory methods rely on complex deterministic and mathematical engineering models. One of the most widely used models, the inverted pendulum, is the source of several algorithms, including those in [14–17]. The Zero Moment Point (ZMP) is the traditional method accepted as an indicator of dynamic stability in bipedal robots [18]. The robot's dynamic balance is preserved when it reaches the ZMP because the its foot's response to the ground balances out the dynamics brought on by its locomotion [18]. The robot's Center of Mass (CoM) is calculated to acquire the ZMP. Therefore, it is necessary to use simulation computations or force sensors that are mounted to the robot's feet. However, for a small-size humanoid robot with constrained system resources and processing rates, calculating the ZMP for each step takes some time. Moreover, traditional approaches mainly rely on dynamics and mathematical models for both the robot and terrain. Therefore, it requires a vast amount of time and effort for designers. The model needs to be redesigned when either the terrain or the kind of robot changes. In addition, the past knowledge and expertise of the designers, who cannot fully explore the robot's potential, also impact the performance of the human-designed model. Traditional control systems' fundamental

drawback is their heavy dependence on the correctness of a mathematical model, which may be affected by joint friction, ground contact force, or other uncertainties.

The RL-based control approach is an alternate method of addressing the issues mentioned above with humanoid robot walking. Due to its adaptable learning capabilities, the RL is increasingly applied in the field of bipedal robotic gait control. The RL is a branch of machine learning that may be used to train complicated control systems without using models. An agent (robot) may learn how to control itself in various circumstances by interacting with the environment, thanks to the RL. According to the environmental states and the agent's behaviors, the environment is modeled in real life to either reward or penalize the agent. The agent works on developing the ability to utilize the past to forecast which behaviors will result in the greatest reward in the future. The development of numerous action policy learning algorithms, many of which are based on the Markov Decision Process (MDP), has been crucial to the model-free learning of bipedal walking in particular [19,20]. The RL aids in getting over the difficulties of dynamic design and computation. For walking problems, the RL is a type of intelligent learning technique. The ZMP location can be controlled using RL approaches to ensure walking stability. While motion control problems can be handled with RL with great performance, legged robot gait control is still difficult because of its complexity.

For the walking pattern, trajectories are generated that allow the robot to walk as desired. Numerous walking parameters are involved in trajectory generation. The procedure of manually adjusting the walking parameters is quite difficult and complex, especially for a sophisticated robot with more than 10 DoF. Only little distortions are permitted by the controls used to construct typical trajectories. Walking might break down due to even the smallest shift. It is possible that these controllers cannot be adjusted to diverse terrains such as slopes and stairs. As a result, it becomes necessary to alter various walking parameters, which significantly raises the cost and workload.

The feature engineering for traditional RL algorithms comes from observations. It might be difficult to extract features for complicated issues, or there may not be enough data from which to develop a strong model. The bipedal gait is substantially more difficult since it calls for a high-dimensional state and action space and demands careful control of each joint while maintaining stability. Extracting high-level features from data with a wide state space and missing observations is now possible thanks to a more recent technique called Deep Neural Networks (DNNs). Deep Reinforcement Learning (DRL) enables an agent to interact with the environment more complicatedly, thanks to recent developments in DNNs. Some end-to-end DRL techniques teach the robot model using the default reward of a simulator. Such DRL controllers, however, have the potential to produce movements that are unsuitable for robots. Utilizing a robot simulator and prior knowledge is one way to mitigate this scenario. Additionally, for RL without previous knowledge, hyperparameters are frequently sensitive. Applications are constrained by low convergence and ineffective training. In this study, an efficient new framework built on the humanoid robot model Robotis-OP2 combined with the controller that generates the traditional trajectory and DRL is proposed to get over these constraints. Most of the standard machine learning methods are suitable for supervised learning or unsupervised learning methods. The main reason for choosing DRL from the machine learning methods in this paper is its interaction with the environment and the high cost of creating datasets on robot systems. It is ensured that the robot can learn the optimum with the reward/punishment mechanism according to the status information received from the environment with the sensors during walking. In this study, the Artificial Neural Network (ANN) is trained directly with data from the robot, without any ready dataset. Thanks to the proposed framework, the optimum walking parameters of the trajectories generated with the walking pattern generator are obtained with the Dueling Double Deep Q Network (D3QN) [21]. The training of the D3QN is carried out using the robot's multi-sensors in the Webots [22] simulator. After determining optimum gait parameters, a robot posture stabilization system in the sagittal plane is proposed. Experimental studies are performed in the Webots simulation environment and

real environment with the proposed framework and Robotis-OP2's walking algorithm by transferring the controllers to the real robot. Experimental results have shown that the Robotis-OP2 humanoid robot can walk more stably on flat ground both in the simulator and in the real environment with the proposed framework than Robotis-OP2's walking algorithm. In addition, it is the first study in the literature to optimize walking parameters for the stable walking of humanoid robots using DRL.

There are many studies in the literature on bipedal walking using RL. In OpenAI Gym [23], many algorithms are employed to resolve bipedal challenges by applying RL to directly control the joints [24]. However, convergence and performance are degraded when these algorithms are utilized to control the gait of a robot whose dynamics are significantly more complicated. Additionally, a nonhuman gait may also be present in the trained model, which is often unsatisfactory. For RL-based gait controllers, previous knowledge-based training procedures have become a critical operational principle.

Recently, there have been several attempts to incorporate RL into bipedal robotic gait control without computing the mathematical model based on model-free RL frameworks. The pose sequence that enables an NAO robot to travel the most distance in the least amount of time while walking on a level surface without falling was discovered by Gil et al. [25] using Q-learning [26]. Liu et al. [27] used the Policy Gradient (PG) [28], which is one of the RL algorithms, to correct the gait model parameters of the NAO humanoid robot to make the gait resistant to unknown disturbances. Lin et al. [29] proposed a method for dynamic bipedal gait and balance control using Q-learning without prior knowledge of the dynamic model. The bipedal robot was able to maintain static stability thanks to the balancing learning approach, which shifts the ZMP on the robot's soles using the movement of the arm and leg. The seesaw and bipedal walking on a level surface were subjected to balancing algorithms. According to the simulation results, the robot might learn to enhance its walking speed behavior using the proposed strategy.

Silva et al. [30] proposed a Q-learning-based method for learning the action policy that enables a robot to walk upright on a slightly inclined surface. The system's proposed design combines a standard gait generator with an RL component on two layers. This situation allows an accelerometer to be used when the slope of the ground the robot is walking on changes to provide a gait adjustment. Experimental studies on a real robot have shown that the stability problem can be successfully solved. Silva et al. [31] aimed to optimize the parameter values of the gait model generator to provide a fast and dynamically stable gait for the DARwIn-OP humanoid robot. They achieved this by using the Temporal Difference (TD) [32] algorithm from the RL methods.

In a two-dimensional simulator, robot walking was accomplished using the Deep Deterministic Policy Gradient (DDPG) [33] technique by Kumar et al. [34]. In around 25,000 episodes, their agents received the target score. The Recurrent Deep Deterministic Policy Gradient (RDDPG) [35] algorithm was used by Song et al. [36] to address the partial observability problem of bipedal walking, and the results were superior to those of the original DDPG approach. A modular framework was presented by Kasaei et al. [37] to provide stable bipedal mobility by tightly linking the analytical walking method with the DRL. To identify the optimal parameters and learn how to increase the robot's stability by modifying the height of the CoM, a learning framework based on evolutionary algorithms and Proximal Policy Optimization (PPO) [38] was designed.

In [39], a controller combined with conventional control and RL was trained to walk the Robotis-OP3 humanoid robot in the PyBullet simulation environment. The study is divided into two parts: pose optimization and DRL. In the first part, they used the RL algorithm, Q-learning, to obtain a combination of the walking parameters of the traditional controller and optimized it according to the robot's state. The PPO algorithm was used to train the multi-layer neural network in the second part, where the action space was created with the first phase. Thus, they obtained which pose order would enable the bipedal robot to perform the required task.

Zhang et al. [40] proposed the LORM (Learn and Outperform the Reference Motion) method, a DRL-based framework for the bipedal robot gait, utilizing prior knowledge of reference motion. In comparison to the reference motion and other motion-based approaches, the agent trained using the proposed method performed better. The proposed method has been validated on the DARwIn-OP humanoid robot in the Webots simulator. Christiano et al. [41] compared the possible trajectories of humans and humanoid robots. They developed a reward function using the data, then improved it with RL and updated the technique to DRL.

A two-level hierarchical control system was used by DeepLoco [42]. Low-level controllers first learned to work on a predetermined time scale and achieve stable walking. Second, by requesting the required step objectives from the low-level controller, the high-level controllers learned the time-scale policy of the steps. The actor-critic DRL method was applied to both levels of the hierarchy similarly. The NAO humanoid robot's stability was maintained while walking on static and moving platforms in the simulated environment utilizing a new hybrid RL framework that was introduced in the study [43]. An iterative actor-critic RL system was used by Wawrzynski [44] to change a humanoid robot's initial slow walk into a quick and capable one.

Two different forms of impedance controllers, to which RL algorithms were applied, were used by Feirstein et al. [45] to the enable limit-loop walking of a straightforward bipedal gait model. Leng et al. [46] proposed a Mean-Asynchronous Advantage Actor-Critic (M-A3C) RL algorithm to directly obtain the robot's final gait without introducing the reference gait. It has been shown that the proposed method can provide continuous and stable gait planning for the bipedal robot. To solve the emerging problems of traditional gait control methods, the DRL algorithm was used in [47,48]. A biped controller based on the DDPG algorithm was created by Liu et al. [49] that can keep stability against static and dynamic disturbances.

A novel reward-adaptive RL technique for bipedal movement was proposed by Huang et al. [50]. They ensured that the control policy was optimized by more than one criterion simultaneously using a dynamic mechanism. The proposed approach used a multi-layered critic to identify a unique value function for each reward component, resulting in hybrid policy gradients. To construct a feedback system that can handle the walking pattern problem, a feedback system that combines an Adaptive Neural Fuzzy Inference System (ANFIS) [51] and a Double Deep Q-network (DDQN) [52] was proposed in [53]. To update the walking parameters, the output of the ANFIS was utilized for training a predictive model called DDQN.

Wong et al. [54] designed an oscillator-based gait model with sinusoidal functions for generating trajectory planning and obtaining bipedal motion for a humanoid robot. To ensure that the robot walks straight, the direction of rotation is considered a parameter of the walking pattern. They used Q-learning to create a simple walking pattern. According to the findings of experiments, the proposed framework enables the humanoid robot to walk steadily and straighter.

The rest of the paper is organized as follows. Section 2 includes the walking pattern generator created for gait planning and inverse kinematics analysis to determine the angles that the robot's leg joints should take with this pattern. To find the optimal walking parameters, calculations of the orientation angles and ZMPs that form the state space of the DRL structure using multi-sensors are given in Section 3. In Section 4, the proposed stable walking framework is detailed. In this section, the optimization system of the walking parameters with the D3QN architecture and the stabilization system of the robot pose in the sagittal plane are analyzed, respectively. Experimental results are given in Section 5. The paper is concluded with the conclusions and future work in Section 6.

## 2. Walking Pattern Generator

In this study, a walking pattern generator is created as given in the flowchart in Figure 1. After determining the ankle and hip trajectories, the angle trajectories of the leg joints of Robotis-OP2 are obtained using inverse kinematic analysis.

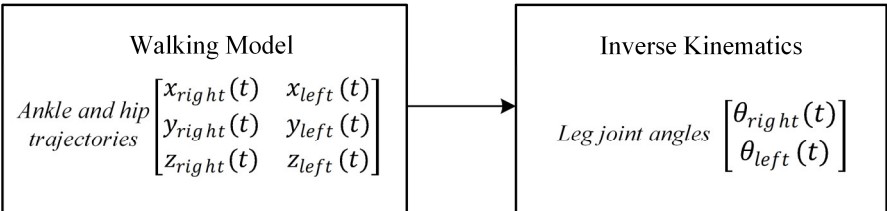

**Figure 1.** Walking pattern generator.

The Robotis-OP2 humanoid robot with 20 DoF weighing 3 kg and its joint placement are shown in Figure 2. The robot's joints are all actuated by the high-torque Dynamixel MX-28T smart servo motor.

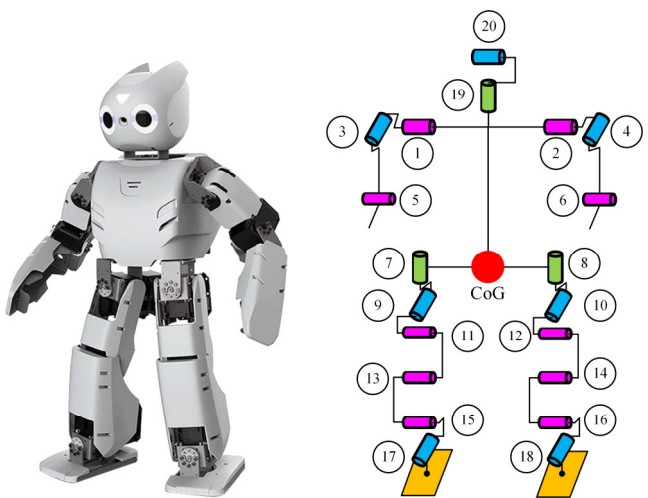

**Figure 2.** Robotis-OP2 humanoid robot and joint placement.

An eight-phase walking cycle is planned to perform the basic walking process, as shown in Figure 3. The first two phases are the beginning phases of walking and include the necessary preparations for trajectory planning.

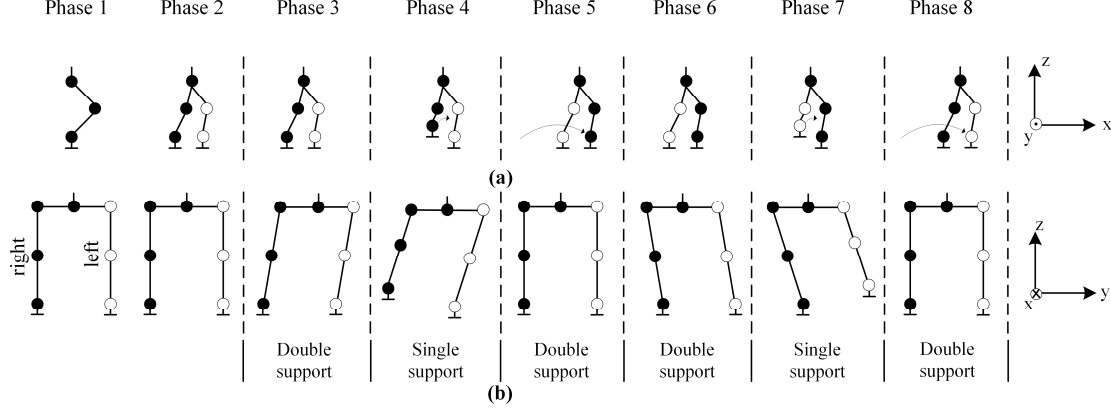

**Figure 3.** Eight-phase walking process: (**a**) sagittal plane, (**b**) frontal plane.

Phase 1: It is the initial phase used to define the robot's trajectory coordinates.

Phase 2: It is the preparation phase of the robot before it starts walking. In this phase, the robot is ready to walk by moving the left foot forward a certain quantity and the right foot a certain quantity back.

Phase 3: This is the phase where the robot starts walking. The robot moves its CoG to the left foot. At this time, the feet are fixed on the ground and only the hip joints are shifted to the left.

Phase 4: When the CoG is moved to the left foot, the right foot begins to step.

Phase 5: The right foot finishes stepping and returns to the support area, and the CoG is ready to move to the right foot.

Phase 6: The robot moves its CoG to the right foot. In the meantime, the feet are fixed on the ground and only the hip joints are shifted to the right.

Phase 7: When the CoG is moved to the right foot, the left foot begins to step.

Phase 8: The left foot finishes stepping and returns to the support area.

Considering these eight phases, the six phases between Phase 3 and Phase 8 are repeated continuously to obtain a continuous walking process. A walking trajectory is planned for this process.

*2.1. Walking Trajectory*

In the gait model, cycloid curve functions are used to generate the ankle and hip trajectories. The walking pattern is generated separately in the x–z planes for the ankle and x–y planes for the hip. The ankle joints are the starting point of trajectory planning, and the hip joints are the ending point of trajectory planning.

2.1.1. Ankle Trajectory

The swing trajectory of the right ankle joint in the x–z plane is given in Equations (1)–(3) for Phase 3, Phase 4, and Phase 5, respectively. The swing trajectory, including these three phases for the left ankle joint, is given in Equation (4).

$$\begin{cases} x_{r\_a}(t) = 0 \\ y_{r\_a}(t) = 0 \, , \, 0 \le t \le \dfrac{t_d}{2} \\ z_{r\_a}(t) = 0 \end{cases} \tag{1}$$

$$\begin{cases} x_{r\_a}(t) = \dfrac{s}{\pi}\left( 4\pi \dfrac{t - \frac{t_d}{2}}{T - 2t_d} - \sin\left( 4\pi \dfrac{t - \frac{t_d}{2}}{T - 2t_d} \right) \right) \\ y_{r\_a}(t) = 0 \\ z_{r\_a}(t) = \dfrac{h}{2}\left( 1 - \cos\left( 4\pi \dfrac{t - \frac{t_d}{2}}{T - 2t_d} \right) \right) \end{cases} , \, \dfrac{t_d}{2} < t \le \dfrac{T - t_d}{2} \tag{2}$$

$$\begin{cases} x_{r\_a}(t) = 2s \\ y_{r\_a}(t) = 0 \, , \, \dfrac{T - t_d}{2} < t \le \dfrac{T}{2} \\ z_{r\_a}(t) = 0 \end{cases} \tag{3}$$

$$\begin{cases} x_{l\_a}(t) = s \\ y_{l\_a}(t) = 0 \, , \, 0 \le t \le \dfrac{T}{2} \\ z_{l\_a}(t) = 0 \end{cases} \tag{4}$$

where t is the current time (s), $t_d$ is the time of shift of the CoG in double support in Phase 5 to Phase 6 and Phase 8 to Phase 3 (s), s is one step length (mm), h is the maximum foot lift height (mm), and T is the walking cycle (Phase 3–8) time (s). In addition, $x_{r\_a}$, $y_{r\_a}$, $z_{r\_a}$ and $x_{l\_a}$, $y_{l\_a}$, $z_{l\_a}$ (mm) represent the trajectories of the right and left ankle joints in the x, y, and, z axes, respectively.

$t_d$ is calculated by Equation (5).

$$t_d = \dfrac{T}{2} \times \text{DSR} \tag{5}$$

where DSR is the double support ratio.

The swing trajectory for the right ankle joint, which includes Phase 6, Phase 7, and Phase 8, is given in Equation (6). The swing trajectory of the left ankle joint is given in Equations (7)–(9) for Phase 6, Phase 7, and Phase 8, respectively.

$$\begin{cases} x_{r\_a}(t) = 2s \\ y_{r\_a}(t) = 0 \;, \; \dfrac{T}{2} < t \le T \\ z_{r\_a}(t) = 0 \end{cases} \tag{6}$$

$$\begin{cases} x_{l\_a}(t) = s \\ y_{l\_a}(t) = 0 \;, \; \dfrac{T}{2} < t \le \dfrac{T + t_d}{2} \\ z_{l\_a}(t) = 0 \end{cases} \tag{7}$$

$$\begin{cases} x_{l\_a}(t) = \dfrac{s}{\pi}\left(4\pi\dfrac{t-\frac{T}{2}-\frac{t_d}{2}}{T-2t_d} - \sin\left(4\pi\dfrac{t-\frac{T}{2}-\frac{t_d}{2}}{T-2t_d}\right)\right) + s \\ y_{l\_a}(t) = 0 \qquad\qquad\qquad , \; \dfrac{T+t_d}{2} < t \le T - \dfrac{t_d}{2} \\ z_{l\_a}(t) = \dfrac{h}{2}\left(1 - \cos\left(4\pi\dfrac{t-\frac{T}{2}-\frac{t_d}{2}}{T-2t_d}\right)\right) \end{cases} \tag{8}$$

$$\begin{cases} x_{l\_a}(t) = 3s \\ y_{l\_a}(t) = 0 \;, \; T - \dfrac{t_d}{2} < t \le T \\ z_{l\_a}(t) = 0 \end{cases} \tag{9}$$

### 2.1.2. Hip Trajectory

The swing trajectories of the right and left hip joints in the x–y plane are given in Equations (10) and (11) for Phase 3, Phase 4, and Phase 5, respectively.

$$\begin{cases} x_{r\_h}(t) = \dfrac{s}{2\pi}\left(4\pi\dfrac{t}{T} - \sin\left(4\pi\dfrac{t}{T}\right)\right) + \dfrac{s}{2} \\ y_{r\_h}(t) = -\dfrac{w}{2}\left(1 - \cos 4\pi\dfrac{t}{T}\right) \quad , \; 0 \le t \le \dfrac{T}{2} \\ z_{r\_h}(t) = h_{ah} - h_b \end{cases} \tag{10}$$

$$\begin{cases} x_{l\_h}(t) = \dfrac{s}{2\pi}\left(4\pi\dfrac{t}{T} - \sin\left(4\pi\dfrac{t}{T}\right)\right) + \dfrac{s}{2} \\ y_{l\_h}(t) = -\dfrac{w}{2}\left(1 - \cos 4\pi\dfrac{t}{T}\right) \quad , \; 0 \le t \le \dfrac{T}{2} \\ z_{l\_h}(t) = h_{ah} - h_b \end{cases} \tag{11}$$

The swing trajectories of the right and left hip joints are given in Equations (12) and (13) for Phase 6, Phase 7, and Phase 8, respectively.

$$\begin{cases} x_{r\_h}(t) = \dfrac{s}{2\pi}\left(4\pi\dfrac{t-\frac{T}{2}}{T} - \sin\left(4\pi\dfrac{t-\frac{T}{2}}{T}\right)\right) + \dfrac{3s}{2} \\ y_{r\_h}(t) = \dfrac{w}{2}\left(1 - \cos 4\pi\dfrac{t-\frac{T}{2}}{T}\right) \qquad , \; \dfrac{T}{2} < t \le T \\ z_{r\_h}(t) = h_{ah} - h_b \end{cases} \tag{12}$$

$$\begin{cases} x_{l\_h}(t) = \dfrac{s}{2\pi}\left(4\pi\dfrac{t-\frac{T}{2}}{T} - \sin\left(4\pi\dfrac{t-\frac{T}{2}}{T}\right)\right) + \dfrac{3s}{2} \\ y_{l\_h}(t) = \dfrac{w}{2}\left(1 - \cos 4\pi\dfrac{t-\frac{T}{2}}{T}\right) \qquad , \; \dfrac{T}{2} < t \le T \\ z_{l\_h}(t) = h_{ah} - h_b \end{cases} \tag{13}$$

where $h_{ah}$ is the height between the ankle joint and hip joint (mm), $h_b$ is the bending height used to adjust the ground clearance of the hip joint (mm), and w is the maximum right-to-left translation when the hip is swinging (mm). In addition, $x_{r\_h}$, $y_{r\_h}$, $z_{r\_h}$ and $x_{l\_h}$, $y_{l\_h}$, $z_{l\_h}$ represent the trajectories of the right and left hip joints in the x, y, and z axes, respectively.

Walking parameters for the half gait cycle (Phase 3, Phase 4, and Phase 5) are shown in Figure 4. As can be seen from the equations generated for the walking trajectory and

Figure 4, the gait model can be adjusted by changing three length parameters (s, h, w), a time parameter (T), and a rate parameter (DSR).

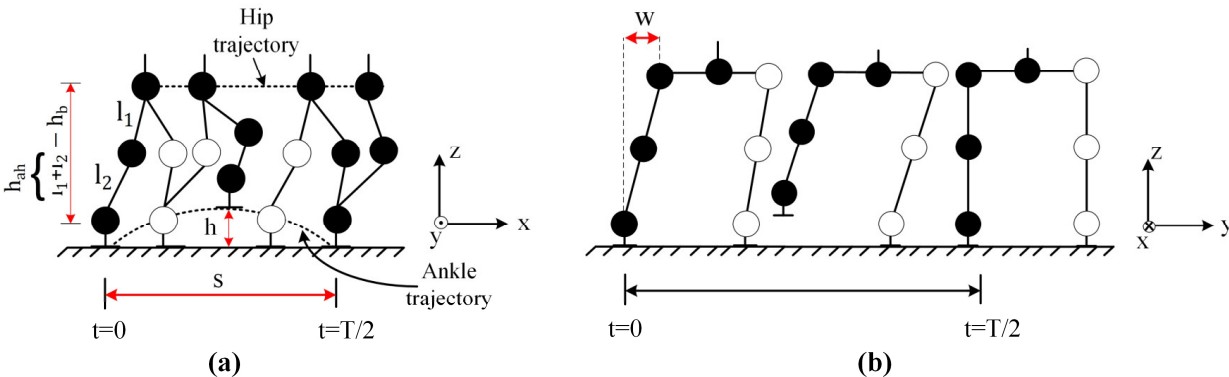

**Figure 4.** Walking parameters for half gait cycle: (**a**) sagittal plane, (**b**) frontal plane.

### 2.2. Kinematic Analysis

### 2.2.1. Forward Kinematics

For the forward kinematics, firstly, the rotation axes of the robot leg are determined, and axes are placed on the joints. The yaw joint in the hip of the robot is not included in the kinematic analysis as it does not affect the walking trajectory for a straight walking task. The lengths of the leg links and placement of the axes are shown in Figure 5.

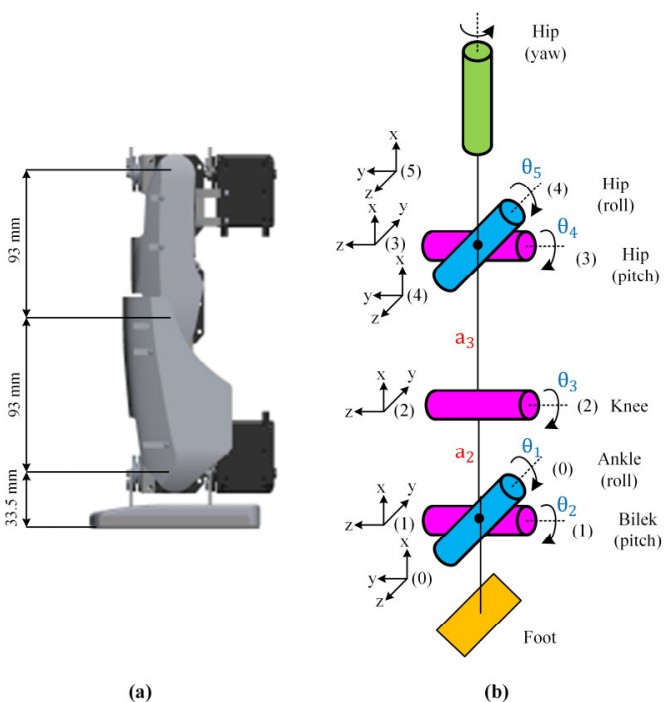

**Figure 5.** Leg joints of Robotis-OP2: (**a**) link lengths, (**b**) placement of the axes.

In Figure 6, the representation of the leg joint angles is given.

The D-H table for the leg of the Robotis-OP2 robot is created as given in Table 1.

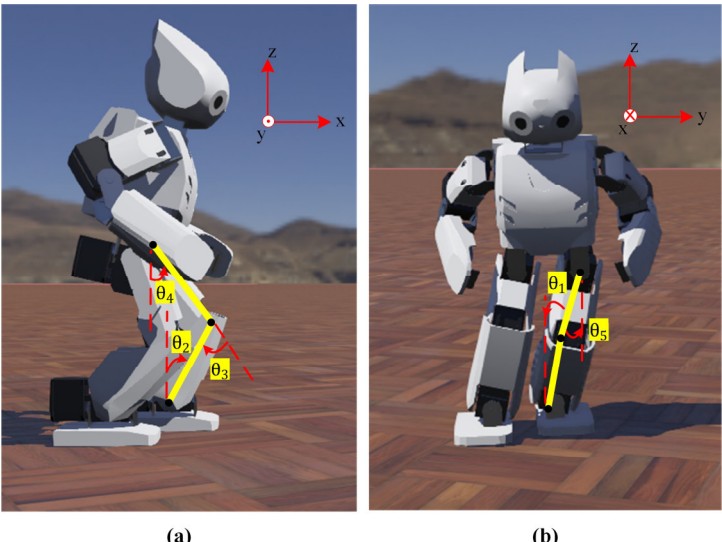

**Figure 6.** Representation of leg joint angles: (**a**) pitch angles, (**b**) roll angles.

**Table 1.** D-H table of Robotis-OP2 robot leg.

| Axis | $\theta$ (rad) | d (mm) | a (mm) | $\alpha$ (rad) |
|------|------|------|------|------|
| 1 | $\theta_1$ | 0 | 0 | $-\pi/2$ |
| 2 | $\theta_2$ | 0 | 93 | 0 |
| 3 | $\theta_3$ | 0 | 93 | 0 |
| 4 | $\theta_4$ | 0 | 0 | $\pi/2$ |
| 5 | $\theta_5$ | 0 | 0 | 0 |

Homogeneous transformation matrices (Equation (A1)) of the leg, which determine the orientation and position of a joint relative to the previous joint, are obtained by using the parameters in the D-H table given in Table 1.

The transformation matrix according to the ankle joint, where the main coordinate system of the Robotis-OP2 hip joint is located, is obtained as in Equation (A2).

In the $T_0^5$ matrix given in Equation (A2), $(n, o, a)_{x,y,z}$ represents the orientation of the hip joint, which indicates the angle of rotation of one coordinate system with relation to another coordinate system, and $p_x$, $p_y$, $p_z$ represents the x, y, and z position coordinates of the hip joint relative to the ankle joint, respectively.

Equation (A3) is obtained using the homogeneous transformation matrices previously obtained.

$p_x$, $p_y$, and $p_z$ are obtained as in Equations (14)–(16), respectively, and forward kinematic analysis is completed.

$$p_x = \cos\theta_1(93\cos\theta_2 + 93\cos\theta_2\cos\theta_3 - 93\sin\theta_2\sin\theta_3) \tag{14}$$

$$p_y = \sin\theta_1(93\cos\theta_2 + 93\cos\theta_2\cos\theta_3 - 93\sin\theta_2\sin\theta_3) \tag{15}$$

$$p_z = -93\sin\theta_2 - 93\cos\theta_2\sin\theta_3 - 93\cos\theta_3\sin\theta_2 \tag{16}$$

### 2.2.2. Inverse Kinematics

When the leg structure of Robotis-OP2 is examined, the first three axes cause the displacement of the hip joint, which is the endpoint, while the last two axes change the rotation of the hip joint. Since the equations to be obtained by inverse kinematics may have more than one solution set, the angle limits are chosen considering the joint angle limits of the Robotis-OP2 legs [55] in Table 2.

**Table 2.** Joint angle limits of Robotis-OP2 legs [55].

| Joint | Limits of Joint Angle (Degree) | | | | | |
| --- | --- | --- | --- | --- | --- | --- |
| | Right Leg | | Left Leg | | Selected Limits | |
| | *min.* | *max.* | *min.* | *max.* | *min.* | *max.* |
| Ankle (roll) $\theta_1$ | $-39$ | 60 | $-58$ | 34 | $-58$ | 34 |
| Ankle (pitch) $\theta_2$ | $-71$ | 78 | $-79$ | 70 | $-79$ | 70 |
| Knee $\theta_3$ | 0 | 128 | $-129$ | 0 | 0 | 128 |
| Hip (pitch) $\theta_4$ | $-101$ | 25 | $-28$ | 96 | $-96$ | 25 |
| Hip (roll) $\theta_5$ | $-57$ | 58 | $-57$ | 53 | $-57$ | 53 |

For inverse kinematic analysis, firstly, both sides of the equation in Equation (A2) are multiplied first by $A_1^{-1}$, and then by $A_2^{-1}A_1^{-1}$, and $\theta_1$, $\theta_3$, and $\theta_2$ are obtained, respectively.

$\theta_1$ is calculated by Equation (17).

$$
\begin{cases}
\theta_1 = \tan^{-1}\left(\frac{p_y}{p_x}\right) \text{ if within the limits of the angle} \\
\theta_1 = \tan^{-1}\left(-\frac{p_y}{p_x}\right) \text{ else}
\end{cases}
\tag{17}
$$

$\theta_3$ is calculated as in Equation (20) using Equations (18) and (19).

$$
\cos\theta_3 = \frac{p_z^2 + (p_x\cos\theta_1 + p_y\sin\theta_1)^2 - 93^2 - 93^2}{2(93)^2}
\tag{18}
$$

$$
\sin\theta_3 = \sqrt{1 - \cos^2\theta_3}
\tag{19}
$$

$$
\begin{cases}
\theta_3 = \tan^{-1}\left(\frac{\sin\theta_3}{\cos\theta_3}\right) \text{ if within the limits of the angle} \\
\theta_3 = \tan^{-1}\left(\frac{-\sin\theta_3}{\cos\theta_3}\right) \text{ else}
\end{cases}
\tag{20}
$$

$\theta_2$ is calculated as in Equation (23) using Equations (21) and (22).

$$
\sin\theta_2 = \frac{-(p_x\cos\theta_1 + p_y\sin\theta_1)93\sin\theta_3 - (93 + 93\cos\theta_3)p_z}{(-93\sin\theta_3)^2 - (93 + 93\cos\theta_3)^2}
\tag{21}
$$

$$
\cos\theta_2 = \frac{-p_z93\sin\theta_3 + \left(p_x\cos\theta_1 + p_y\sin\theta_1\right)(93 + 93\cos\theta_3)}{(93 + 93\cos\theta_3)^2 + (-93\sin\theta_3)^2}
\tag{22}
$$

$$
\begin{cases}
\theta_2 = \tan^{-1}\left(\frac{\sin\theta_2}{\cos\theta_2}\right) \text{ if within the limits of the angle} \\
\theta_2 = \tan^{-1}\left(-\frac{\sin\theta_2}{\cos\theta_2}\right) \text{ else}
\end{cases}
\tag{23}
$$

The equation in Equation (A2) is multiplied by $A_3^{-1} A_2^{-1} A_1^{-1}$. The rotation matrix representing pitch used in obtaining $\theta_4$ and $\theta_5$ angles is given in Equation (24).

$$
R(\theta) = \begin{bmatrix} \cos\theta & 0 & \sin\theta \\ 0 & 1 & 0 \\ -\sin\theta & 0 & \cos\theta \end{bmatrix} = \begin{bmatrix} n_x & o_x & a_x \\ n_y & o_y & a_y \\ n_z & o_z & a_z \end{bmatrix}
\tag{24}
$$

$\theta_4$ is calculated as in Equation (25).

$$
\begin{cases}
\theta_4 = \tan^{-1}\left(\frac{\sin\theta_3(n_z\sin\theta_2 - \cos\theta_2(n_x\cos\theta_1 + n_y\sin\theta_1)) - \cos\theta_3(n_z\cos\theta_2 + \sin\theta_2(n_x\cos\theta_1 + n_y\sin\theta_1))}{\cos\theta_3(-n_z\sin\theta_2 + \cos\theta_2(n_x\cos\theta_1 + n_y\sin\theta_1)) - \sin\theta_3(n_z\cos\theta_2 + \sin\theta_2(n_x\cos\theta_1 + n_y\sin\theta_1))}\right) \text{ if within the limits of the angle} \\
\theta_4 = \tan^{-1}\left(-\frac{\sin\theta_3(n_z\sin\theta_2 - \cos\theta_2(n_x\cos\theta_1 + n_y\sin\theta_1)) - \cos\theta_3(n_z\cos\theta_2 + \sin\theta_2(n_x\cos\theta_1 + n_y\sin\theta_1))}{\cos\theta_3(-n_z\sin\theta_2 + \cos\theta_2(n_x\cos\theta_1 + n_y\sin\theta_1)) - \sin\theta_3(n_z\cos\theta_2 + \sin\theta_2(n_x\cos\theta_1 + n_y\sin\theta_1))}\right) \text{ else}
\end{cases}
\tag{25}
$$

$\theta_5$ is calculated as in Equation (26).

$$\begin{cases} \theta_5 = \tan^{-1}\left(\frac{n_y\cos\theta_1 - n_x\sin\theta_1}{o_y\cos\theta_1 - o_x\sin\theta_1}\right) \; if \; within \; the \; limits \; of \; the \; angle \\ \theta_5 = \tan^{-1}\left(-\frac{n_y\cos\theta_1 - n_x\sin\theta_1}{o_y\cos\theta_1 - o_x\sin\theta_1}\right) else \end{cases} \tag{26}$$

With the rotation matrix in Equation (24), the θ value, which is the robot's body tilt angle in the sagittal plane, is adjusted. The θ shown in Figure 7 is equal to the sum of the pitch angles of the leg ($\theta_2 + \theta_3 + \theta_4$) and can be selected at desired values so as not to cause the robot to fall while walking.

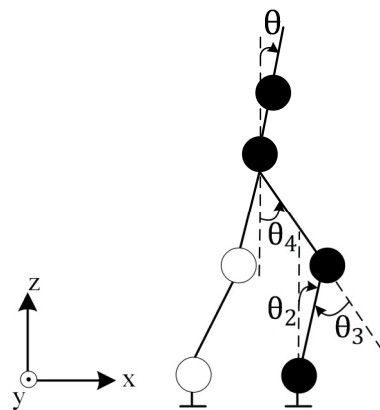

**Figure 7.** Representation of robot's body tilt angle in the sagittal plane.

Using the generated walking trajectory, the position coordinates $p_x$, $p_y$, and $p_z$ of the hip joint relative to the ankle joint are calculated for both legs. The calculation operations consist of subtracting the ankle joint's position from the position of the hip joint at each time step.

## 3. Orientation Angles and Gait Stability Criterion

With an Inertial Measurement Unit (IMU), the three-dimensional orientation of an object in space can be found [56]. The ZMP is where the sum of weight and vertical inertial forces equals zero. When the robot maintains at least one flat foot on the ground while walking and has active ankle joints, the ZMP may be employed as a gait stability criterion.

### 3.1. Calculation of Orientation Angles

The gyroscope and accelerometer are embedded in the CM-740 controller board, which is the sub-controller of Robotis-OP2. Raw accelerometer and gyroscope data are read from the controller board. The IMU consists of a three-axis ADXL335 accelerometer with an analog output and a three-axis LYPR540AH gyroscope with an analog output.

Considering Robotis-OP2's straight gait in the sagittal plane, the body tilt angle is the pitch angle. Regarding the coordinate framework, the positive pitch angle is counterclockwise, and the positive roll angle is clockwise. Figure 8 shows the pitch and roll angles of the body.

The gyroscope measures the angular velocities $\omega_x$, $\omega_y$, and $\omega_z$ along the x, y, and z axes, respectively, while the accelerometer measures the linear accelerations $a_x$, $a_y$, and $a_z$ along the x, y, and z axes, respectively. However, the data from the sensors are raw data. These data are values read from the 10-bit ADC unit of the CM-740 sub-controller.

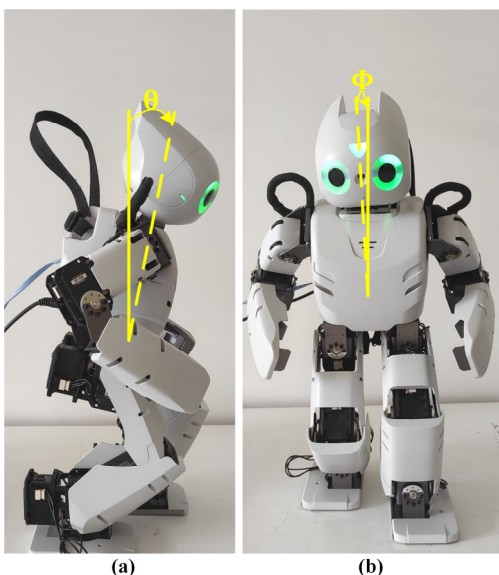

**Figure 8.** Body tilt angles: (**a**) pitch angle, (**b**) roll angle.

Table 3 shows the parameters of the accelerometer, which can measure acceleration in the range of ± 3g in three axes, and the gyroscope, which can measure angular velocity in the range of ±1600 degrees/s in three axes.

**Table 3.** Parameters of accelerometer and gyroscope.

| Parameter | Accelerometer | Gyroscope |
|---|---|---|
| $ADC_{value}$ | 0~1023 | 0~1023 |
| $V_{ref}$ | 3.3 V | 3.3 V |
| Bit Rate | 1024 | 1024 |
| $V_{zero}$ | 1.65 V | 1.65 V |
| Sensitivity | 0.33 V/g | 0.0008 V/(degree/s) |
| R | a | $\omega$ |

The raw data from the accelerometer and gyroscope are converted into physical units expressing g (9.8 m/s$^2$) in terms of the magnitude of the gravity for the accelerometer and degrees/s for the gyroscope using the equation in Equation (27).

$$R = \left( \frac{ADC_{value} \times V_{ref}}{BitRate} - V_{zero} \right) \times \frac{1}{Sensitivity} \tag{27}$$

The acceleration values are converted to pitch ($\theta$) and roll ($\Phi$) angles by Equation (A4). The pitch ($\theta$) and roll ($\Phi$) angles are obtained using angular velocity values by Equation (A5).

Although noisy in the short term, the accelerometer offers reliable data over the long run. Angle shifts happen over lengthy time scales, but the gyroscope is stable and delivers precise information on changing orientation in the short term. To calculate the pitch and roll angles of the humanoid robot body properly, accelerometer and gyroscope data from the IMU are combined to eliminate errors. While the Kalman filter [57] has several parameters that need to be adjusted, the complementary filter is easier to set up as the complementary filter has only one filter parameter that needs to be adjusted. However, in our experimental studies on the complementary filter and the Kalman filter, the Kalman filter gives more accurate results than the complementary filter under dynamic effects. For this reason, the use of the Kalman filter, which is widely preferred for IMUs, is used as a sensor fusion method, and is preferred in this study.

### 3.2. Calculation of Gait Stability Criterion

In bipedal gait analysis using force or pressure measurements, the Center of Pressure (CoP) [58] is also frequently utilized. Figure 9a illustrates how the system is stable and the foot's CoP corresponds with the ZMP when the ZMP is inside the support polygon in the single support phase. When the instantaneous ZMP in Figure 9b is at or beyond the support polygon, it means that an imbalanced moment M has arisen and cannot be corrected by the foot response forces R. In this paper, it is proposed to create a stable walking framework with higher stability by using the ZMP of the robot to take into account the dynamic balance.

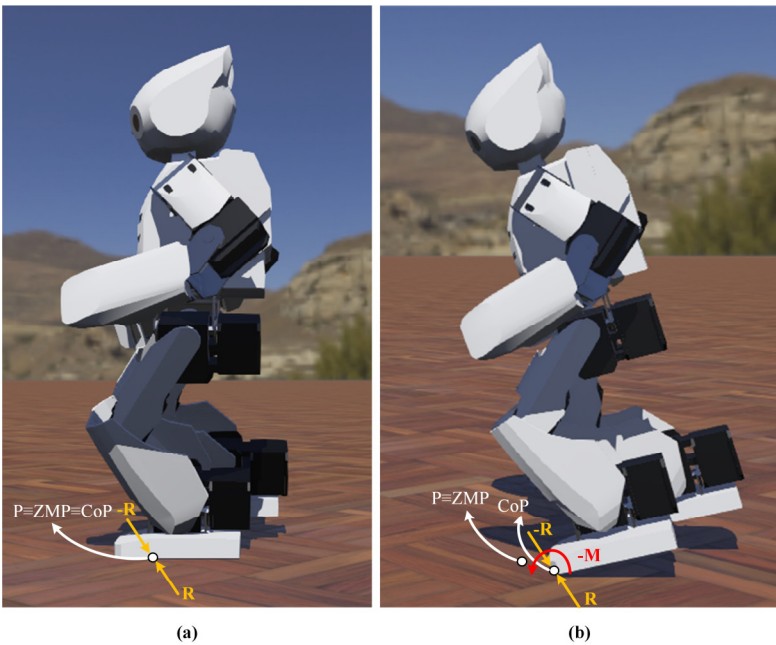

**Figure 9.** Relationship between ZMP and CoP: (**a**) dynamically stable gait, (**b**) dynamically unsteady gait.

Force Sensitive Resistors (FSRs) can be used to calculate the ZMP by providing measurements of the robot's vertical ground reaction force and the moment of the ankle joint about its axis of rotation. In this paper, the feet of the Robotis-OP2 humanoid robot are replaced with the FSR set.

The placement of FSRs on the sole in Robotis-OP2 and the numbering of forces are given in Figure 10a. The starting point is defined in the coordinate system (0,0) of the ZMP regarding the center of the left ankle. In the Webots simulation environment, ZMPs are placed according to their dimensions as in Figure 10b.

When the instantaneous measurements of the force values measured with FSRs are examined, it is seen that they have noise. Since the mass of the robot is 3 kg, the sum of the force values measured from the FSRs should be equal to the approx. weight force of the robot, 29.43 N, from the formula W = mg. Since the force values measured from the FSRs have noise, it has been seen that their sum is higher than it should be. In this paper, a one-dimensional Kalman filter [59], frequently used in scientific literature, is applied to remove the noise in the measured force values. The one-dimensional Kalman filter is a state estimation method based on a single measurement parameter.

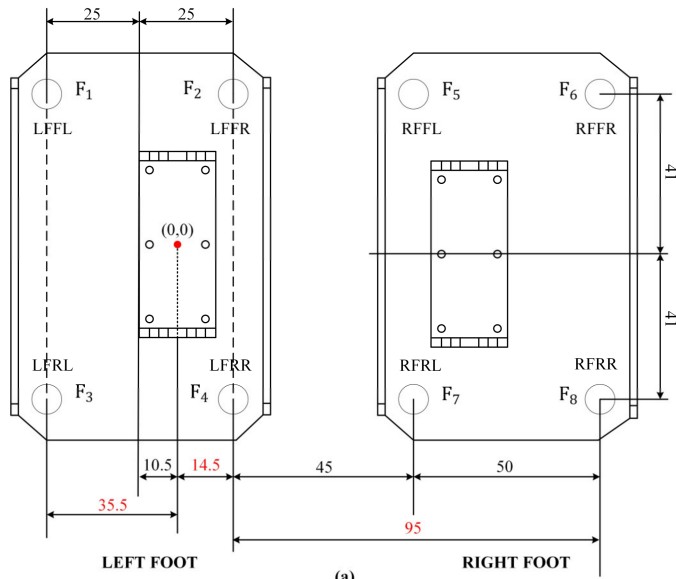

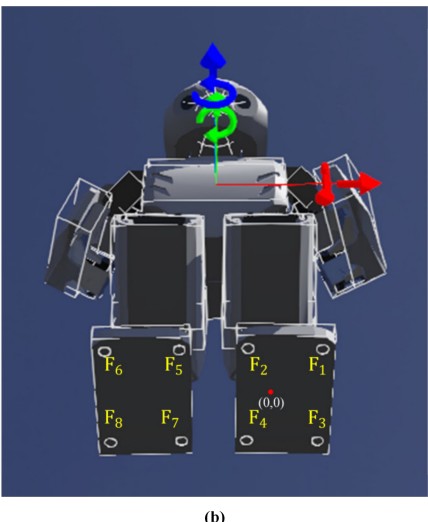

**Figure 10.** Robotis-OP2's FSRs: (**a**) layout and numbering of forces (top view), (**b**) placement in Webots.

For the walking trajectory, the forward–backward direction represents the x-axis, and the right–left direction represents the y-axis. Therefore, the ZMP coordinates are calculated for the x and y axes. During a gait that includes single and double support phases, there are three statuses of the right and left feet relative to each other. These are statuses where the feet are on a line, the left foot is forward, and the right foot is forward. When Figure 10a is examined, the distances of the FSRs to each other on the y-axis remain constant during walking. Therefore, the y-axis coordinate of the ZMP ($ZMP_y$) is calculated by the Equation (28) in all three statuses.

$$ZMP_y = \frac{59.5(F_5 + F_7) + 109.5(F_6 + F_8) + 14.5(F_2 + F_4) - 35.5(F_1 + F_3)}{\sum_{i=1}^{8} F_i} \qquad (28)$$

The x-axis coordinate of the ZMP ($ZMP_x$) is calculated by the equation in Equation (29) when the feet are on a line.

$$ZMP_x = \frac{41(F_1 + F_2) + 41(F_5 + F_6) - 41(F_3 + F_4) - 41(F_7 + F_8)}{\sum_{i=1}^{8} F_i} \qquad (29)$$

$\text{ZMP}_x$ is calculated by the equation in Equation (30) by being affected by translation as $x_{left} - x_{right}$ when the left foot is forward.

$$\text{ZMP}_x = \frac{41(F_1+F_2)+\left(41-(x_{left}-x_{right})\right)(F_5+F_6)-41(F_3+F_4)-\left(41+(x_{left}-x_{right})\right)(F_7+F_8)}{\sum_{i=1}^{8}F_i} \tag{30}$$

$\text{ZMP}_x$ is calculated by the equation in Equation (31) by being affected by translation as $x_{right} - x_{left}$ when the right foot is forward.

$$\text{ZMP}_x = \frac{41(F_1+F_2)+\left(41+(x_{right}-x_{left})\right)(F_5+F_6)-41(F_3+F_4)-\left(41-(x_{right}-x_{left})\right)(F_7+F_8)}{\sum_{i=1}^{8}F_i} \tag{31}$$

## 4. Proposed Framework of Gait Parameter Optimization

The main purpose of this paper is to ensure that the Robotis-OP2 humanoid robot walks stably without falling. The most important task for this purpose is to obtain the most suitable gait parameters of the trajectories generated with the walking pattern generator. The proposed framework for stable walking is given in Figure 11.

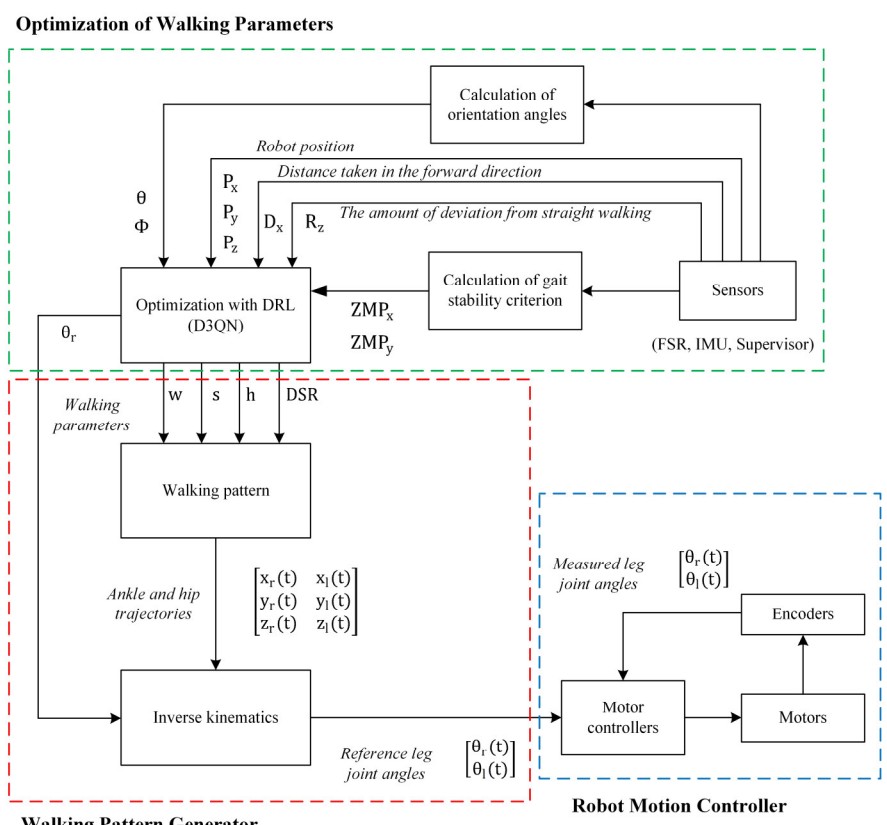

**Figure 11.** Proposed framework for stable walking.

D3QN, one of the DRL algorithms, is used for the training of walking parameters to perform the straight walking process stably.

There are six walking parameters ($h_b$, $w$, $s$, $h$, $T$, DSR) in the hip and ankle trajectories generated for bipedal walking. Using any DRL algorithm to optimize all parameters makes convergence very difficult as the size of the action space is very large. Therefore, the action space should be simplified after determining the values that the parameters can take, taking into account the kinematic structure of the robot. The limits of the walking parameters shown in Figure 12 are approximately determined as a result of kinematic calculations and observations in Webots.

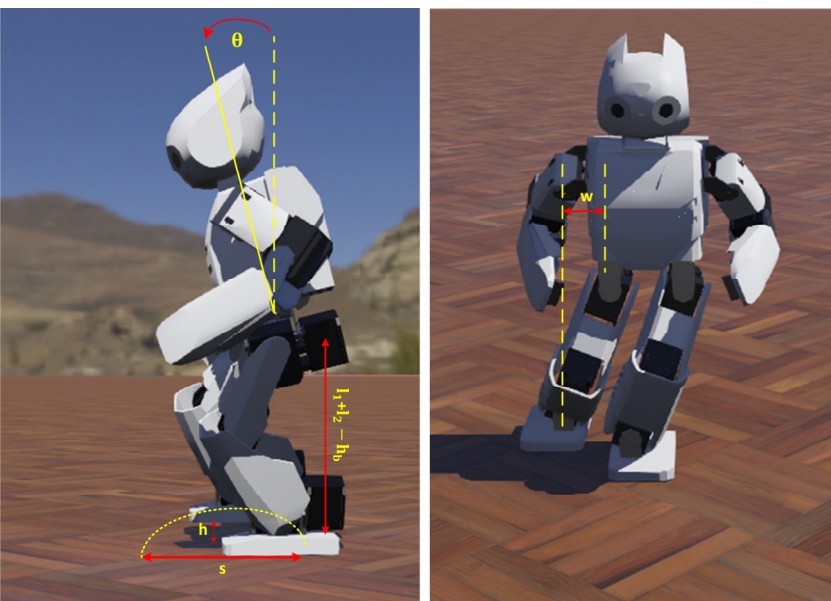

**Figure 12.** Representation of walking parameters.

When the robot walks steadily, the CoM height is constant, as seen from the hip trajectory. However, it can be in a negligible small range of variation due to noise during movement. To decrease the robot's energy usage without bending the knee joint too much, $h_b = 25$ mm was determined as a constant. In addition, gravity creates unwanted torques on the robot's joints. The effect of gravity becomes more explicit when the robot switches between a double support and a single support during walking. The robot weight, supported by feet, is suddenly supported only by the support foot. In this paper's scope, emphasis is placed on the double support phase in the eight-phase gait cycle generated for a stable gait. The DSR parameter cannot, therefore, be selected too low. The walking speed must be low so that the inertia forces are negligible in the gait cycle, which is created similarly to quasi-static walking. Therefore, the walking cycle time was chosen as T = 2 s.

*4.1. Training with Dueling Double Deep Q Network*

First of all, methods such as the DQN and DDQN are examined in our studies, and the direct use of the Dueling Double Deep Q Network (D3QN) is adopted without wasting time to overcome the problem of overestimating the state values of these methods. The D3QN has two improvements to solve the problem of state value overestimation. First, the D3QN uses a dual network structure that uses a $Q_1$ evaluation network to select an action for the next step with $a_{max} = \text{argmax}_a Q_1(s', a; \theta)$. Then, $a_{max}$ is evaluated with Equation (32) by a target network $Q_2$ to reduce the overestimation.

$$\begin{cases} a_{max} = \text{argmax}_a Q_1(s', a; \theta) \\ y = r + \gamma Q_2(s', a_{max}; \theta^-) \end{cases} \tag{32}$$

Second, the D3QN divides the action-value function $Q_\pi(s, a)$ into two parts as in Equation (33): a state-value function $V_\pi(s)$ related to state and an advantage function $A_\pi(s, a)$ regarding both state and action.

$$Q_\pi(s, a; \theta, \alpha, \beta) = V_\pi(s; \theta, \alpha) + A_\pi(s, a; \theta, \beta) \tag{33}$$

where $\theta$ is the common parameter of $V_\pi$ and $A_\pi(s, a)$, $\alpha$ and $\beta$ are the parameters of $V_\pi(s)$ and $A_\pi(s, a)$, respectively.

The D3QN structure used is shown in Figure 13.

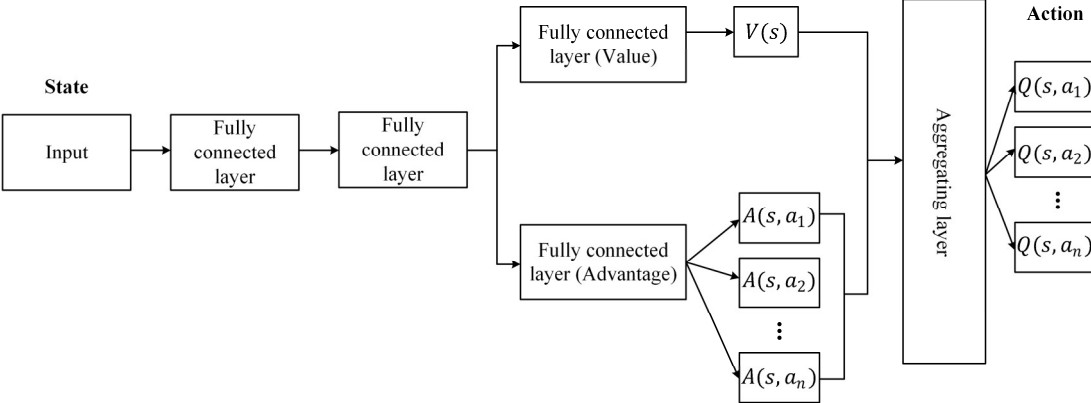

**Figure 13.** Structure of D3QN.

For stable humanoid walking implementation, the D3QN learns the optimal policy by performing Algorithm 1.

---

**Algorithm 1 D3QN**

---

Initialize a stable humanoid gait system (Environment)
Initialize $\mathcal{D}$ repeat buffer with $M$ capacity
Initialize $\mathcal{Q}_1$ Evaluation Network with random $\theta$ parameters
Initialize $\mathcal{Q}_2$ Target Network with $\theta^- \leftarrow \theta$ parameters
Refill buffer with data generated by random policy
**while** *True* **do**
  Restart stable humanoid walking system
  **while** *No termination* **do**
    Choose a random action $a$ with probability $\epsilon$, otherwise choose action
$a = \underset{a}{\mathrm{argmax}} Q_1(s_t, a; \theta)$
    Perform action $a$, get reward $r$ and get next state $s'$
    Save experience tuple $\langle s, a, r, s' \rangle$ to $\mathcal{D}$
    Sample mini batch from $\mathcal{D}$
    **if** *there is a termination at step $j + 1$* **then**
      $Y_j = r_j$
    **else**
      $a_{max} = \underset{a}{\mathrm{argmax}} Q_1(s', a; \theta)$
      $Y_j = \gamma \underset{a'}{\max} Q_2(s', a_{max}; \theta^-)$
    $\hat{Y}_j = Q_1(s, a; \theta)$
    Minimize loss to train $Q_1$ for updating $\theta$
    Update Target Network at every $C$ steps: $\theta^- \leftarrow \theta$
  **if** $Q_1$ converges **then**
    $Q_2 = Q_1$
    $Q_{optimal} = Q_2$
    **break**

---

### 4.1.1. The Architecture of D3QN

The layers, the number of neurons, and the activation functions of the D3QN, whose structure is given in Figure 13, are given in Table 4. ANN structures with hidden layer neurons such as 128, 1024, and 2048 are tried, but the best result is obtained from the ANN designed with the number of neurons in Table 4.

**Table 4.** The layer structure of D3QN.

| Layer Name | Layer Type | Neuron Numbers | Activation Type |
|---|---|---|---|
| Input | Fully connected | 9 | - |
| Common FC | Fully connected | 512 | ReLU |
| Common FC | Fully connected | 256 | ReLU |
| FC1 for Value | Fully connected | 128 | ReLU |
| FC1 for Advantage | Fully connected | 128 | ReLU |
| FC2 for Value | Fully connected | 1 | Linear |
| FC2 for Advantage | Fully connected | 14,400 | Linear |
| Output | Fully connected | 14,400 | - |

The training phase includes 30 thousand episodes using the Adam optimizer, and each episode consists of seven steps. The D3QN algorithm was created in such a way that the robot's walking for 6T time was taken into account for each training step. In each step performed in the episode, the robot takes an action and performs its movement. If the robot can take 12 steps without falling, it receives a reward and the next step of the episode is passed from where it left off. However, if the robot falls at any time during the walking or has an inverse kinematics calculation error, it cannot receive a reward and is punished directly. Next, the environment is reset, and the next episode begins.

The hyperparameters selected for training the D3QN are listed in Table 5.

**Table 5.** The layer structure of D3QN. The selected hyperparameters for D3QN.

| Hyperparameter | Value |
|---|---|
| Mini batch size | 32 |
| Replay memory size | 15,000 |
| Discount factor $\gamma$ | 0.95 |
| Learning rate | 0.0001 |
| Initial exploration rate $\epsilon$ Min. exploration rate $\epsilon_{min}$ | 1 0.01 |

### 4.1.2. State Space

The state space consists of a nine-dimensional array containing states that are important for the stable walking task as given in Table 6.

**Table 6.** State space.

| State | Unit | Size |
|---|---|---|
| Orientation angles (pitch ($\theta$ ) and roll ($\Phi$)) | deg. | 2 |
| Position ($P_x$, $P_y$, $P_z$) | mm | 3 |
| Rotation in the $z - axis$ ($R_z$) | rad. | 1 |
| Zero Moment Point ($ZMP_x$ and $ZMP_y$) | mm | 2 |
| Distance taken in the forward direction ($D_x$) | mm | 1 |

where $\theta$ and $\Phi$ are the pitch and roll angles of the robot body calculated using the Kalman filter, respectively; $P_x$, $P_y$, and $P_z$ are the positions of the robot in the x, y, and z axes, respectively; $R_z$ is the rotation of the robot in the z-axis; $ZMP_x$ and $ZMP_y$ are the ZMPs in the x and y axes, respectively; and $D_x$ is the forward displacement distance of the robot.

### 4.1.3. Action Space

In the rotation matrix of inverse kinematics equations, the range of $\theta_r$, which determines the pitch angle $\theta$ of the robot, shown in Figure 12, is not selected at high values to be close to an upright walk. $\theta_r$ is added to the action space as the fifth walking parameter. The ranges of the five walking parameters are discretized as in Table 7, which can be meaningful for stable walking.

**Table 7.** Discrete values of parameters in the action space.

| Parameter | Discrete Values |
|:---:|:---:|
| w | [25 30 35 40 45 50 55 60 65 70] |
| s | [25 30 35 40 45 50 55 60 65 70 75 80] |
| h | [25 30 35 40 45 50] |
| DSR | [0.3 0.4 0.5 0.6] |
| $\theta_r$ | [−3 −8 −13 −18 −23] |

As can be seen in Table 7, the action space has a total dimension of $10 \times 12 \times 6 \times 4 \times 5 = 14{,}400$ consisting of w, s, h, DSR, and $\theta_r$ combinations.

### 4.1.4. Reward Function

R used to optimize walking parameters is expressed by the reward function (34).

$$R = 10D_x + 3S_d + 6S_b - 5S_h \tag{34}$$

where $D_x$ is the distance the robot takes in the forward direction (mm), $S_d$ is the counter that determines the scale of the robot in walking straight without deviating (the counter increases as the robot walks under the given threshold deviation ($\pm 0.01$ rad)), $S_b$ is the counter that expresses the robot's stable walking (the counter increases when the ZMP on the x and y axes is within the support polygon of the robot), and $S_h$ is the counter that measures the error in the ZMP calculation (when the robot's feet do not touch the ground due to disruptive effects, ZMP cannot be calculated because information cannot be obtained from the FSRs).

The reward is determined as −100 when the robot falls or when an inverse kinematics calculation error is encountered during the robot's movement.

### 4.2. Body Posture Balancing

Depending on the circumstance, maintaining balance after a walking impairment can be a challenging procedure, including many solutions. Three fundamental strategies—the ankle, hip, and stepping strategies—have been identified as a result of experimental research on this process in humans. These three balancing strategies, shown in Figure 14, are also adopted in humanoid robots. In this paper, a balancing method consisting of a PID controller is proposed so that the Robotis-OP2 robot can cope with robotic uncertainties and walk while maintaining balance. The use of the hip strategy is preferred for balancing. The main reason for choosing the hip strategy is that it directly affects body orientation, as the hip is the connecting piece between the legs and the body. In addition, it is more resistant to disruptive effects than the ankle strategy, and the angle of inclination of the body can be controlled with a single parameter because the hip is chosen as the end joint when creating kinematic equations for walking.

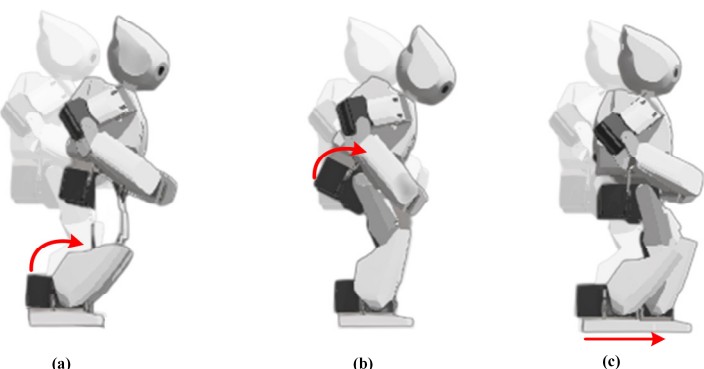

(a)  (b)  (c)

**Figure 14.** Balancing strategies: (**a**) ankle, (**b**) hip, (**c**) step.

The studies have been conducted on a real-time closed-loop control of the Robotis-OP2's pitch orientation for body stabilization control during walking. The block diagram of the proposed structure based on the feedback of the IMU is given in Figure 15.

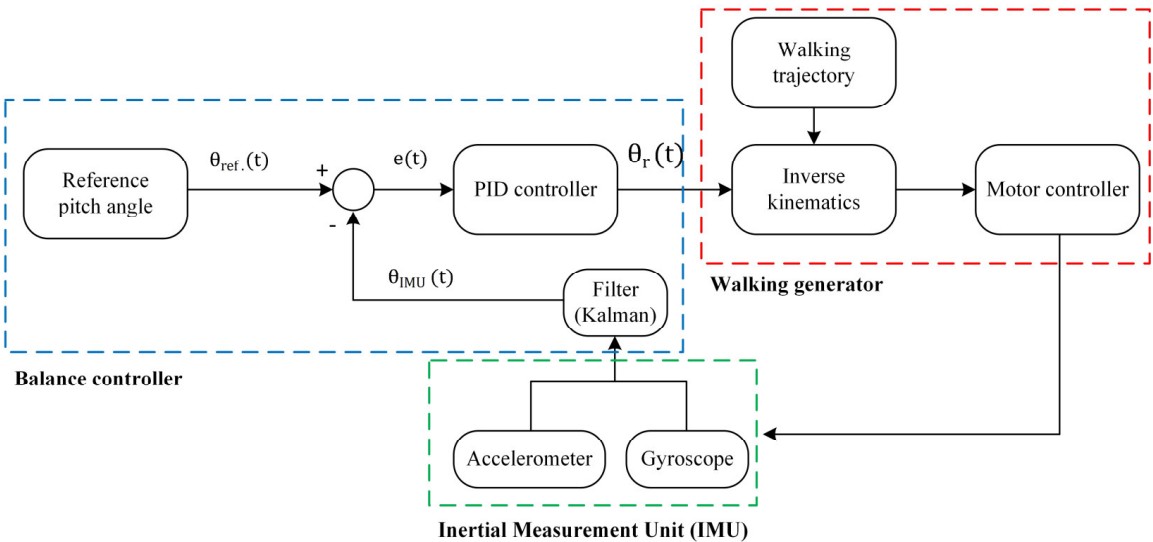

**Figure 15.** Block diagram of body stabilization control during walking.

### 5. Experimental Results

First, experimental studies were carried out to determine the parameters used to calculate the orientation angles and gait stability criterion. The accelerometer, bias, and measurement noise variances of the Kalman filter were determined as 0.001, 0.003, and 0.03, respectively, by taking a time step of 16 ms. Initial values of the process noise, measurement noise, and estimated error variances of the one-dimensional Kalman filter were determined as 0.001, 0.25, and 1, respectively. Moreover, by default, only the P controller with a gain coefficient $K_p$ of 10 is used for the position control of the servo motor in the Webots. In the experimental studies, proportional gain $K_p$ and integral gain $K_i$ coefficients were determined as 30 and 0.9, respectively, by trial and error.

As shown in Figure 16, experimental studies were carried out on a desktop computer with NVIDIA GTX Titan X Pascal GPU, Intel i5 4th generation 3.4 GHz processor, and 8 GB RAM. The training of the D3QN was carried out in the Webots simulator with the controller written using Tensorflow 2.2.0, Keras 2.3.1, and Python 3.8 versions by utilizing the power of parallel computing thanks to the CUDA support of the GPU.

During the training process, the rewards received in each episode were recorded. Weights were stored in every 100 episodes. After the training was completed, the average reward obtained by taking the average of the last 40 episodes was drawn, and it was observed that a highly fluctuating graph appeared. This situation is because training includes too many episodes. The graph obtained by taking the average of the last 200 episodes instead of the last 40 episodes is obtained in Figure 17.

Desktop Computer

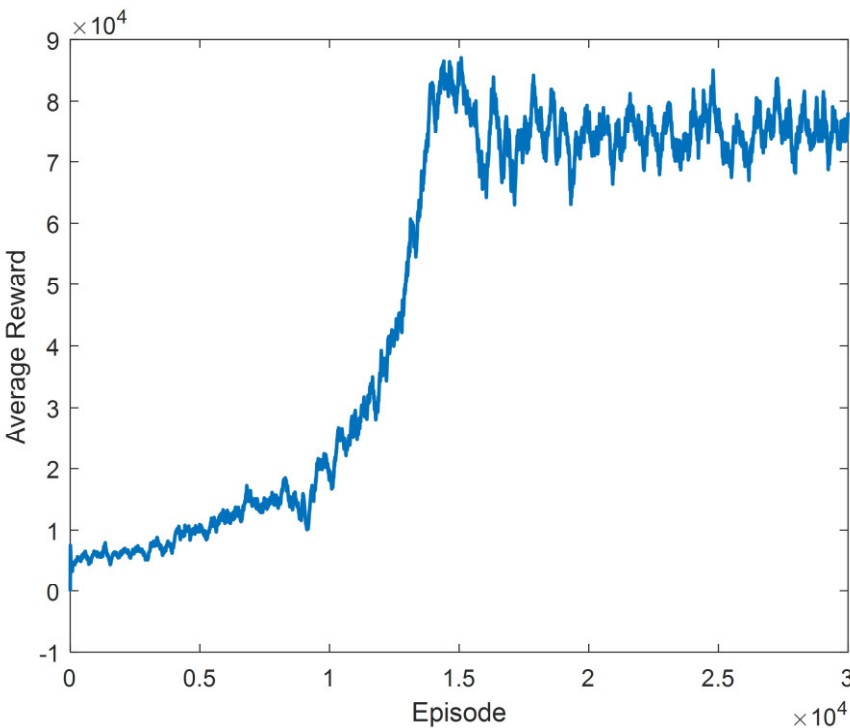

**Figure 16.** Schematic diagram of the training process with DRL.

**Figure 17.** Average reward graph with an average of last 200 episodes.

When the training results in Figure 17 are examined, it is seen that the average reward increases exponentially up to the roughly 16,000th episode. Afterwards, the average reward remained within a specific range. During the training, it has been observed that it converges to the parameters that can perform many successful walking processes. However, action sequences consisting of the actions in Table 8 were obtained repeatedly in at least the last 9000 episodes.

**Table 8.** Optimal gait parameters obtained as a result of training.

| Parameter | Value | Unit |
|-----------|-------|------|
| w | 50 | mm |
| s | 60 | mm |
| h | 30 | mm |
| DSR | 0.6 | - |
| $\theta_r$ | −13 | degree |

With the values in Table 8, $D_x$, $S_d$, $S_b$, and $S_h$ for six walking cycles (6T) were obtained as 709.2 mm, 711, 581, and 0, respectively, and the reward value calculated by Equation (34) was obtained as 12,711. The saved weights were loaded and tested in the Webots R2020b simulator. The walking trajectories obtained with the optimized gait parameters were visualized with graphs in MATLAB R2021b for the gait cycle. The three-dimensional representation of the walking model is given in Figure 18.

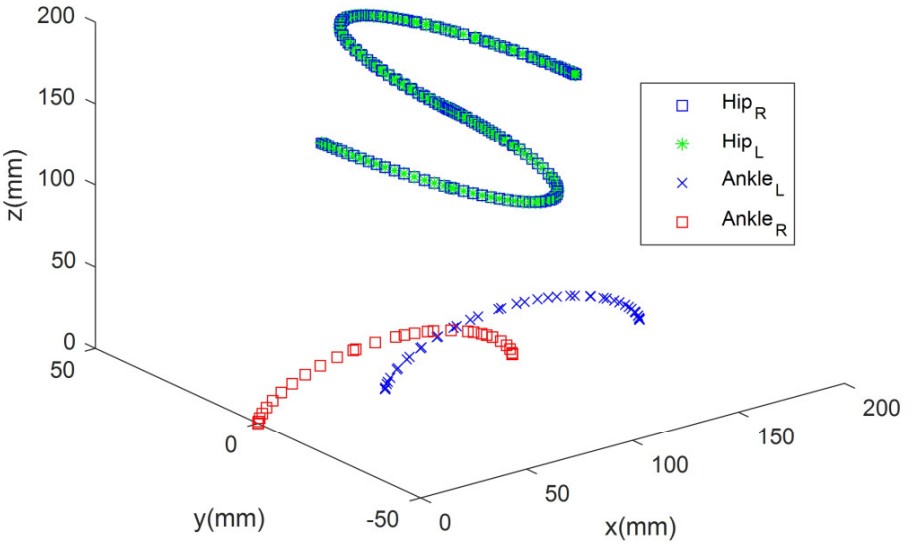

**Figure 18.** Three-dimensional representation of the walking model.

When Figure 18 is examined, the robot's speed is 60 mm/s in the x-axis, and the robot's hip performs a swinging movement of a maximum of 50 mm. In addition, while walking, the robot's foot rises a maximum of 30 mm from the ground.

The ankle and hip joint trajectories for the right and left legs, along with their phases, are shown in Figure 19. The position coordinates of the hip joint relative to the ankle joint $p_x$, $p_y$, and $p_z$ were calculated for both legs by the walking trajectory. The calculation process consists of subtracting the ankle joint's position from the hip joint's position at each time step.

Although there was no disturbance on the ground, the results of straight walking were obtained using the same walking parameters (Table 8) for the four walking cycles with the closed loop. The gain coefficients $K_p$, $K_i$, and $K_d$ obtained by trial and error for the PID controller for body posture balancing were determined as 0.8, 7.1, and 0.01, respectively.

According to Figure 19a, although the hip trajectory in the x-axis is the same for the right and left legs, it has mirrored oscillations for the ankle. According to Figure 19b, the ankle and hip trajectories are the same for both legs. According to Figure 19c, although the hip trajectory is the same for the right and left legs, the ankle trajectory differs.

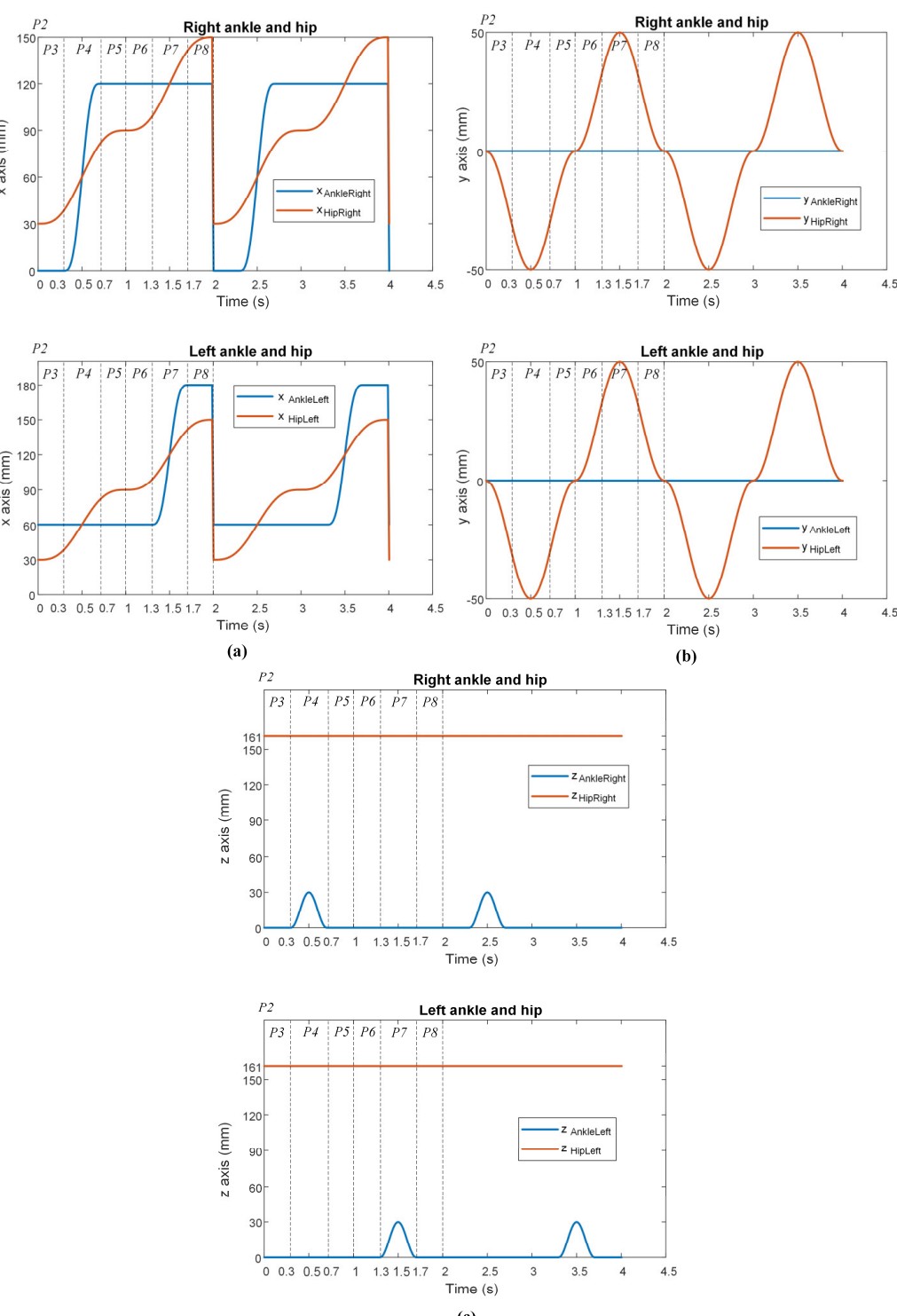

**Figure 19.** Trajectories of the ankle and hip: (**a**) x-axis, (**b**) y-axis, (**c**) z-axis.

The walking algorithm inside the Robotis-OP2 uses a method to construct a gait model based on CPG-based paired oscillators that perform sinusoidal trajectories [60]. Many parameters are available in the algorithm to adjust the gait. However, it has been decided that only a subset of these parameters can be set to facilitate its use by the user [55]. Other parameters are set to default values that are known to work well. The robot's gait can be adjusted if the user wishes by changing the default values stored in the "config.ini" configuration file shown in Figure 20 [55].

```
[Walking Config]
x_offset                 = -10.0;
y_offset                 = 5.0;
z_offset                 = 20.0;
roll_offset              = 0.0;
pitch_offset             = 0.0;
yaw_offset               = 0.0;
hip_pitch_offset         = 13.0;
period_time              = 600.0;
dsp_ratio                = 0.1;
step_forward_back_ratio  = 0.28;
foot_height              = 40.0;
swing_right_left         = 20.0;
swing_top_down           = 5.0;
pelvis_offset            = 3.0;
arm_swing_gain           = 1.5;
balance_knee_gain        = 0.3;
balance_ankle_pitch_gain = 0.9;
balance_hip_roll_gain    = 0.0;
balance_ankle_roll_gain  = 0.0;

[Robot Config]
time_step                = 16.0;
camera_width             = 320.0;
camera_height            = 240.0;
```

**Figure 20.** Default parameters in the "config.ini" file [55].

The results for walking obtained with the proposed framework and Robotis-OP2's walking algorithm [60] are compared as in Figures 21 and 22.

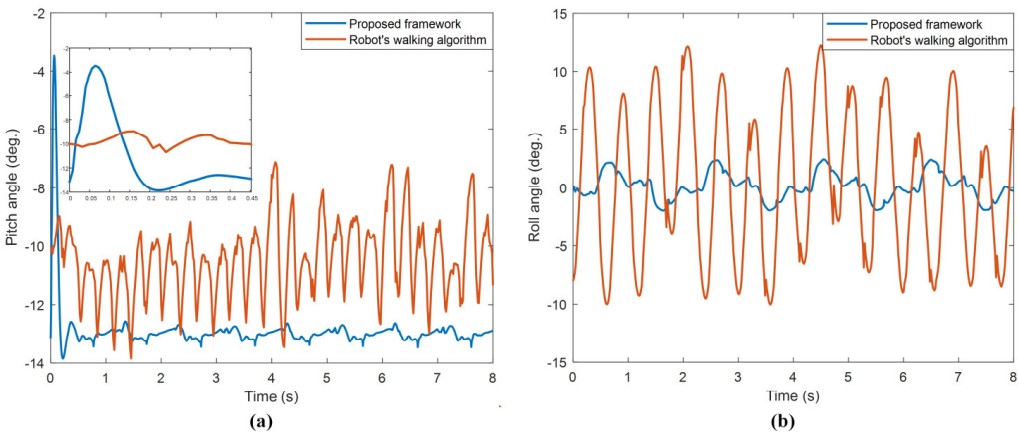

**Figure 21.** Comparison of orientation angles calculated with Kalman filter: (**a**) pitch angle, (**b**) roll angle.

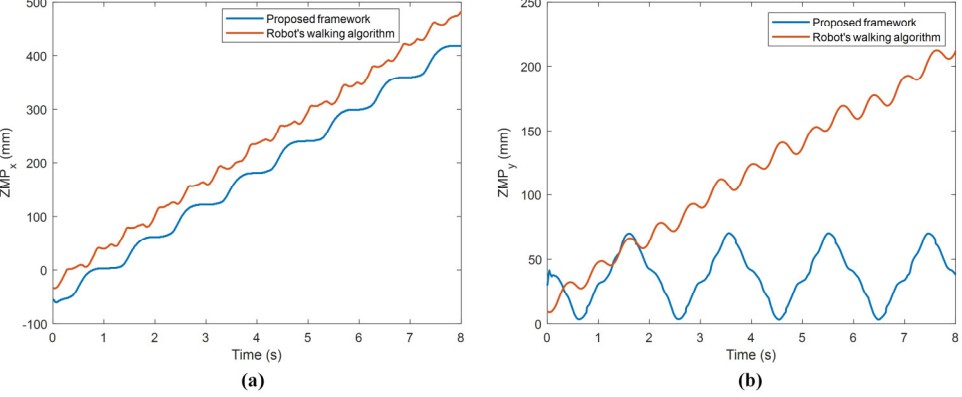

**Figure 22.** Comparison of ZMP values: (**a**) ZMP$_x$, (**b**) ZMP$_y$.

When Figure 21a is observed, at the beginning of the walking, an instantaneous jumping movement occurred in the time it took for the system response with the PID controller to settle on the reference input. However, since this jumping is in the first approx. 0.18 s of walking, it is observed that it did not adversely affect the walking stability of the robot. When the oscillations of the orientation angles in Figure 21 are examined, it is seen that the proposed framework has very little angle change compared to the other. In Table 9, the ranges of the robot orientation angles given in Figure 21 are compared.

**Table 9.** The ranges of orientation angles for two methods.

| Method | Pitch Angle *(deg.)* | | Roll Angle *(deg.)* | |
|---|---|---|---|---|
| | min. | max. | min. | max. |
| Proposed framework | −13.446 | −12.995 | −1.964 | 2.332 |
| Robot's walking algorithm | −13.457 | −7.126 | −10.033 | 12.233 |

With the proposed framework according to Table 9, the robot has very little swinging during walking compared to the robot's walking algorithm for both pitch and roll orientation angles.

When Figure 22a is examined, it is seen that the $ZMP_x$ increases by oscillating more smoothly with the proposed frame compared to the other. Considering the step length is 60 mm, the fact that the robot's $ZMP_x$ moves on 474 mm in the forward direction during four gait cycles indicates quite stable walking. When Figure 22b is examined, it is seen that, with the proposed framework, $ZMP_y$ oscillates regularly in a certain period, and there is only about 12 mm of vertical misalignment at the end of the walk. The other method deviates considerably from the vertical alignment compared to the proposed framework. These experimental results showed that very stable walking was obtained with the proposed framework. In addition, when all the experimental results were evaluated, it was seen that the proposed framework was quite successful in terms of both the balanced walking and stable walking of the robot. The proposed framework is also very useful for optimizing gait parameters for different state and action spaces of the DRL. The user can optimize the robot's gait using the proposed framework for the robot's desired speed and walking pattern generator.

The controller, which was developed after the experimental studies at Webots, was transferred to the real Robotis-OP2 humanoid robot with the remote control tool located in the robot window of Webots. Figure 23 shows the robot's initial states of walking.

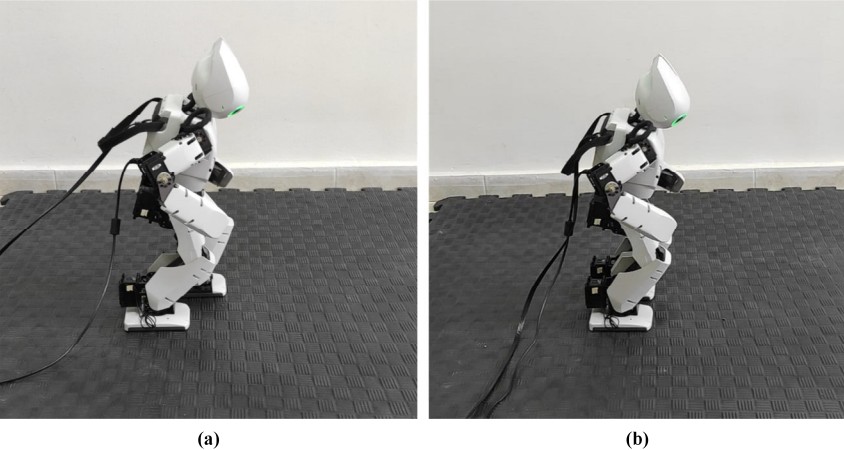

(a)　　　　　　　　　　　　　　(b)

**Figure 23.** The robot's initial states of walking: (**a**) proposed framework, (**b**) walking algorithm of robot.

A comparison of the orientation angles of the walking performed with the proposed framework and Robotis-OP2's walking algorithm is given in Figure 24.

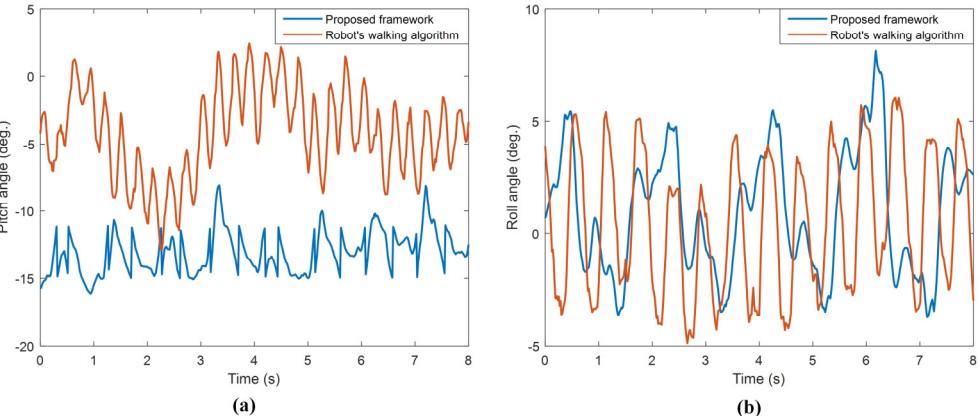

**Figure 24.** Comparison of orientation angles of real robot during walking: (**a**) pitch angle, (**b**) roll angle.

In Figure 24a, it was observed that the oscillation amplitude of the pitch angle was lower. The results in Figure 24b showed that the roll angles of the robot oscillate quite close to each other. Experimental results on the real robot also proved that the robot is more stable during walking.

## 6. Conclusions and Future Work

In this paper, a robust algorithm is developed with the proposed framework for the stable walking of a humanoid robot. The proposed framework consists of a traditional trajectory generator controller and DRL structure. This study is the first study in the literature on the walking of a humanoid robot using the DRL algorithm, D3QN.

Experimental studies were carried out on the Robotis-OP2 humanoid robot in both simulation and real environments. For experimental studies, the walking trajectory of the robot was created using cycloid curves. Leg kinematic analyses were performed to obtain the angles of the leg joints so that the robot could reach the desired trajectories. The data obtained from multi-sensors on the robot were made sense of. The training of the D3QN for the optimization of walking parameters was carried out in the Webots simulator. With the system based on the training of the D3QN, the optimum parameters of the walking trajectory were obtained for stable walking. It converged to the optimum parameters approximately in the 21,000th episode.

For the robot body to maintain its balance in the sagittal plane, a stabilization system was created using the hip strategy and PID controller. Experimental results were obtained by performing walking tests by combining the obtained values and the balancing system. When the robot body was asked to walk with a pitch angle of $-13°$ in the sagittal plane, the robot successfully walked with a swing between $-13.446°$ and $-12.995°$. The body roll angle was between $-1.964°$ and $2.332°$. It was observed that the ZMP values in the x and y axes, which are the stable walking criteria of the robot, also have a very smooth oscillation. For walking where the robot has a step of 60 mm per second, over four walking periods, the $ZMP_x$ moved on 474 mm. Moreover, the $ZMP_y$ shifted only 12 mm from the starting coordinate. These experimental results proved that the robot has a stable gait. Experimental results were also compared with the Robotis-OP2 robot's walking algorithm, and it was seen that the proposed framework had a more stable gait.

Based on the study in this paper, many future studies are planned. Firstly, using the proposed framework in this paper, the robot will be able to walk stably on an unknown inclined ground. The robot's posture will be adjusted in real-time to ensure that the body pitch angle is at the desired reference value as if it were walking on flat ground. Then, studies will be carried out to enable the robot to walk longer with lower energy consumption. Next, more detailed research will be conducted to understand better the natural bipedal

gait mechanisms, including in humans. In addition, it is aimed to develop a gait model for applications to the lower limbs' exoskeleton for both healthy and disabled people.

**Author Contributions:** Conceptualization, Ç.K., A.U. and C.G.; methodology, Ç.K. and A.U.; software, Ç.K. and A.U.; validation, Ç.K. and A.U.; formal analysis, Ç.K. and A.U.; investigation, Ç.K., A.U. and C.G.; resources, Ç.K. and A.U; data curation, Ç.K. and A.U.; writing—original draft preparation, Ç.K. and A.U.; writing—review and editing, Ç.K., A.U. and C.G.; visualization, Ç.K.; supervision, A.U. and C.G. All authors have read and agreed to the published version of the manuscript.

**Funding:** Scientific and Technological Research Council of Turkey (TUBITAK) and NVIDIA.

**Data Availability Statement:** The study did not report any data.

**Acknowledgments:** This study was supported by the Scientific and Technological Research Council of Turkey (TUBITAK) grant numbered 117E589. Additionally, the GTX Titan X Pascal GPU in this research was donated by NVIDIA Corporation.

**Conflicts of Interest:** The authors declare no conflict of interest.

**Appendix A**

$$A_i = \begin{bmatrix} \cos\theta_i & -\sin\theta_i.\cos\alpha_i & \sin\theta_i\sin\alpha_i & a_i\cos\theta_i \\ \sin\theta_i & \cos\theta_i.\cos\alpha_i & -\cos\theta_i\sin\alpha_i & a_i\sin\theta_i \\ 0 & \sin\alpha_i & \cos\alpha_i & d_i \\ 0 & 0 & 0 & 1 \end{bmatrix} \tag{A1}$$

$$T_0^5 = \begin{bmatrix} n_x & o_x & a_x & p_x \\ n_y & o_y & a_y & p_y \\ n_z & o_z & a_z & p_z \\ 0 & 0 & 0 & 1 \end{bmatrix} = A_1 A_2 A_3 A_4 A_5 \tag{A2}$$

$$T_0^5 = \begin{bmatrix} -s_1s_5 - c_1(c_2(c_5s_3s_4 - c_3c_4c_5) + s_2(c_3c_5s_4 + c_4c_5s_3)) & c_1(c_2(s_3s_4s_5 - c_3c_4s_5) + s_2(c_3s_4s_5 + c_4s_3s_5)) - c_5s_1 \\ c_1s_5 - s_1(c_2(c_5s_3s_4 - c_3c_4c_5) + s_2(c_3c_5s_4 + c_4c_5s_3)) & c_1c_5 + s_1(c_2(s_3s_4s_5 - c_3c_4s_5) + s_2(c_3s_4s_5 + c_4s_3s_5)) \\ s_2(c_5s_3s_4 - c_3c_4c_5) - c_2(c_3c_5s_4 + c_4c_5s_3) & c_2(c_3s_4s_5 + c_4s_3s_5) - s_2(s_3s_4s_5 - c_3c_4s_5) \\ 0 & 0 \\ c_1(c_2(c_3s_4 + c_4s_3) + s_2(c_3c_4 - s_3s_4)) & c_1(93c_2 + 93c_2c_3 - 93s_2s_3) \\ s_1(c_2(c_3s_4 + c_4s_3) + s_2(c_3c_4 - s_3s_4)) & s_1(93c_2 + 93c_2c_3 - 93s_2s_3) \\ c_2(c_3c_4 - s_3s_4) - s_2(c_3s_4 + c_4s_3) & -93s_2 - 93c_2s_3 - 93c_3s_2 \\ 0 & 1 \end{bmatrix} \tag{A3}$$

$$\theta_{acc.} = \tan^{-1}\left(\frac{a_y}{\sqrt{a_x{}^2 + a_z{}^2}}\right)\frac{180}{\pi} \; and \; \Phi_{acc.} = \tan^{-1}\left(\frac{a_x}{\sqrt{a_y{}^2 + a_z{}^2}}\right)\frac{180}{\pi} \; (degree) \tag{A4}$$

$$\theta_{gyro.} = \int_0^t \omega_y dt \; and \; \Phi_{gyro.} = \int_0^t \omega_x dt \; (degree) \tag{A5}$$

where dt is the sampling period.

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
