# Peer review of "Development of a New Robust Stable Walking Algorithm for a Humanoid Robot Using Deep Reinforcement Learning with Multi-Sensor Data Fusion"

_electronics, doi:10.3390/electronics12030568_

Round 1
Reviewer 1 Report
This manuscript developed a framework for stable walking of a humanoid robot using the DRL algorithm, D3QN. There is no doubt that the Authors have done a lot of work, and the experimental results also prove the feasibility of the proposed framework. But I think there are two issues that can be discussed:
1) The key innovation of this manuscript can be seen in Figure 12 and Algorithm 1, the Authors adopted a D3QN to optimize walking parameters and it makes the robot walk more stable than using the Robotis-OP2’s walking algorithm. It is still recommended to add comparison methods, especially those based on machine learning, and to discuss how the number of trainable parameters of D3QN can affect the results.
2) There are too many formulas and constants, especially the basic parameters of the robot, which lack universality. It is recommended to give them in detail as an appendix if possible.
Reviewer 2 Report
This paper use deep reinforcement learning with multi-sensor data fusion to development a new algorithm. This paper is technically novel. The effectiveness of the algorithm is verified by a series of simulations and experiment. However, some issues need to be carefully sorted before the paper can be accepted.
1. In 3.1.1, the formula arrangement of the left ankle and right ankle will confuse the reader, please revise the arrangement.
2. The authors should explain and analyze the parameters in Figure 4.
3. The Kalman filter is mentioned at the end of section 4.1, and an explanation should be given as to why this method was chosen.
4. The analysis of experimental results in Chapter 6 of the article is too short to make the reader understand effectively, and more explanation should be given on the experimental results and the purpose of the experiment.
5. The advantages of the technique used in the article over traditional methods could be discussed in more detail.
6. The authors need to check the whole article to remove grammatical errors.
Reviewer 3 Report
Please provide clearer aim of the study in Abstract and Introduction section.
Emphasize novelty and contribution.
Last section can be improved. Please extend further work paragraph taking into consideration usefulness of the described model to:
- improve performance of bipedal robot walk toward lower energy consumption (= longer autonomy),
- better understanding of mechanisms of natural bipedal gait, including in humans,
- enhance model to application within lower limbs exoskeleten for human users (both healthy and disabled).
Please describe if the AI-generated gait can be more effective than natural, human-derived?
Reviewer 4 Report
In my opinion, sections 1 (Introduction) and 2 (Related works) should be merged.
Lines 256-257: “The robot’s joints are all controlled by the high-torque Dynamixel 257 MX-28T smart servo motor” – please reconsider, a motor (even a smart one) can actuate a robot joint. Controlling a joint requires an entire system (motor, sensor/transducer, driver)
Lines 283-284: “In the gait model, cycloid curve functions are used to generate the ankle and hip trajectories.” – this sentence in lines 283-284 is repeated identically in lines 288-289.
Figure 10 has no relevance to the paper.
Lines 467-468: “In this paper, a one-dimensional Kalman filter [59] is 468 applied to remove the noise in the read force values.” – please provide some more explanations.
Lines 609-612: How were the controller gains selected/determined?
Lines 614-616: The explanations given are too brief. At least a logic diagram of the program(s) implementing the controller operation should be presented (same as Algorithm 1 was presented between lines 534-535).
Round 2
Reviewer 4 Report
The authors have taken into account all the problems indicated by the reviewer and made the necessary changes in the paper. I believe that, with the current changes, the paper can be published in the journal.